# Consumer Knowledge and Willingness Pertaining to the Adoption of a Sustainable Diet: A Scoping Review

**DOI:** 10.3390/nu16244254

**Published:** 2024-12-10

**Authors:** Connor Dupuits, Elaine Mooney, Amanda McCloat

**Affiliations:** National Centre of Excellence for Home Economics, School of Home Economics, Atlantic Technological University (ATU), St. Angelas, Sligo F91 C634, Ireland; elaine.mooney@atu.ie (E.M.); amanda.mccloat@atu.ie (A.M.)

**Keywords:** sustainable diet, scoping review, health, consumer knowledge and willingness

## Abstract

The current food system is harming both planetary and human health. The shift to a sustainable diet can help alleviate both adverse effects. The aim of this review was to conduct a scoping review of the literature pertaining to consumer knowledge and willingness concerning the adoption of a sustainable diet. A total of 45 papers met the eligibility criteria. Preferred Reporting Items for Systematic Reviews and Meta-analyses Extension for Scoping Reviews (PRISMA-ScR) guidelines were employed to conduct the scoping review. Studies reported that many participants have misconceptions regarding the meaning of a sustainable diet, and their willingness to adopt a sustainable diet was oftentimes low. Evidence also suggests that the lack of knowledge regarding sustainable diets and the reluctance to reduce meat consumption are chief factors hindering the transition to a sustainable diet. Gender imbalance was also evident with females forming the majority of total participants. During the time of the literature search, no studies were conducted in Ireland. Research needs to be conducted in this country, specifically on young consumers, to explore their knowledge and willingness to adopt a sustainable diet. This would provide further insights into the research area of sustainable diets.

## 1. Introduction

As the global population is projected to approach 10 billion by 2050, ensuring the sustainability of our food systems is more essential than ever [1,2,3]. The current food system is exhausting planetary boundaries, and according to the EAT-Lancet Commission, it is the entity that causes the greatest source of environmental degradation and has the most effect on the Earth system [2] (p. 461). This is evident as the food system expels roughly a third of GHG emissions [4] whilst also contributing to a cascade of detrimental consequences such as biodiversity loss, eutrophication, land change, and deforestation [2,5,6]. As a result, the need for the transition to a sustainable diet is advised to alleviate these harmful consequences caused by the current food system [2,7]. In addition, this strain on the environment is mirrored by increasing public health concerns.

From a global perspective, in 2017, 20% of deaths were associated with a poor diet [8]. Globally, 2 billion adults are overweight, and 670 million of those adults have obesity [1]. Additionally, this figure is expected to rise as 1 billion people are projected to have obesity by 2030 [9]. Conversely, the Food and Agricultural Organization (FAO) and World Health Organization (WHO) [1] noted that “820 million people go to bed hungry every night” (p. 5). More recently, FAO et al. [10] reported that much of the world is still undernourished, stating that between 691 and 783 million people faced hunger in 2022. Thus, displaying a scenario whereby both undernutrition and overnutrition are in sync.

As a consequence, the current food system appears to be creating a triad of adverse effects, namely the destruction of the planet, rising obesity rates, and the prevalence of worldwide hunger. A paper published in The Lancet by Swinburn et al. [11] labelled these effects as “The Global Syndemic” because they “co-occur in time and place, interact with each other to produce complex sequelae, and share common underlying societal drivers” (p. 791). Thus, it appears that to feed a population of almost 10 billion people by 2050, the adoption of a sustainable diet is imperative to mitigate the destructive effects of the current food system on the planet whilst also bettering health.

The interpretation of what constitutes a sustainable diet used throughout this review has been extracted from the FAO, which defines sustainable healthy diets as “dietary patterns that promote all dimensions of individuals’ health and wellbeing; have low environmental pressure and impact; are accessible, affordable, safe and equitable; and are culturally acceptable” [1] (p. 9). Thus, a sustainable diet is one that is not only environmentally adequate and optimal for human health but also cost-effective, accepted by consumers, pragmatic, and feasible.

This paper aims to conduct a scoping review of the literature pertaining to consumer knowledge concerning a sustainable diet and their willingness to adopt a sustainable diet. This will expose research gaps to be further explored.

## 2. Materials and Methods

The review type chosen is a scoping review. A scoping review is “a type of evidence synthesis that aims to systematically identify and map the breadth of evidence available on a particular topic, field, concept, or issue, often irrespective of source” [12] (p. 950). A scoping review was implemented as it charts the current literature available without focusing on precise research questions [13]. Ergo, a scoping review was appropriate for this review as they are useful for identifying and analysing knowledge gaps, examining how this particular field conducts research, and pinpointing the available evidence pertinent to this topic [14]. Thus, aligning with the overall aim of this review.

The guidelines set out in the Preferred Reporting Items for Systematic Reviews and Meta-analyses Extension for Scoping Reviews (PRISMA-ScR) by Tricco et al. [15] were employed to conduct this review. The strategy to locate the articles included in this review can be seen in Figure 1.

Search terms (Table 1) derived from the review’s objective were used as part of the online database advanced search strategy. The databases searched were Web of Science, ScienceDirect, EBSCOHost, GreenFile, EBSCOHost MEDLINE, and PubMed. The searches were conducted in January 2023.

The articles selected were required to meet pre-determined eligibility criteria. To meet the inclusion criteria, articles must have been peer-reviewed, published in the previous 10 years, primary research articles, available in English, and allude to at least one of the following: (1) consumer knowledge of a sustainable diet and (2) consumer willingness to adopt a sustainable diet. Aside from failing to meet the inclusion criteria, articles were excluded if they focused on novel forms of protein, e.g., jellyfish, hybrid products, seaweed, etc. These were excluded because these are specific food products, not diets. Also excluded were reviews, such as systematic reviews, as these are secondary research [16].

To achieve the aim of this review, data were extracted from the included studies, as evident in the results section. Although the results from each article are extracted, a thorough analysis of each article’s results is not presented in this review, as an in-depth analysis of results would be best suited to a systematic review [14,17,18]. However, to fulfil the aim of this review, qualitative content analysis was employed as this is suitable and permitted to be conducted in a scoping review [17]. Moreover, an inductive approach was utilised to collate the article results, which were pivotal to the review’s aim. Likewise, results were coded into either ‘knowledge’ or ‘willingness’.

Database searches yielded 905 articles, which were uploaded to the referencing management software Zotero 6.0.26. These were initially screened for title and abstract per the eligibility criteria. Articles passed initial screening if their title contained words or sentences included in the search terms derived from each concept (“sustainable diet” and “knowledge/willingness”) (Table 1). The abstracts of articles that passed screening by title were then screened in more detail. Articles progressed to the next stage of selection if they met the eligibility criteria aforementioned. Hence, 116 articles were uploaded to Rayyan.

Rayyan is a research collaboration software [19]. Rayyan enables researchers to screen articles based on their eligibility criteria. The invited researchers can select articles they wish to include, exclude, or are undecided about. Any disagreements are highlighted, and the researchers can discuss these to decide whether to include or exclude these articles. The methods and results of these articles were reviewed for eligibility as per inclusion criteria. After examining the articles uploaded to Rayyan, two independent researchers were invited to review the articles, and 45 articles were deemed acceptable to be included in the review (Figure 1). Any disagreements among articles were resolved by the second independent reviewer.

## 3. Results

As depicted in Figure 2, research in this area increases each subsequent year, with 60% (*n* = 27) of the studies included in this review conducted in the three preceding years.

Demographically, the majority (*n* = 28) have been conducted in Europe, with the remaining carried out in North America (*n* = 3), Oceania (*n* = 3), South America (*n* = 2), and Asia (*n* = 3). Multi-country studies (*n* = 6) are not included in the demographic frequency count (Figure 3).

Regarding methods, eight used qualitative methods, thirty-one used quantitative methods, and six used a mixed-methods approach (Figure 4).

The gender of participants involved in many of the studies remained consistent. Male participation was low compared to that of females. Zero studies had over 51% male participants, whereas 36 studies had over 51% female participation. Furthermore, 17 studies consisted of over 60% females. The highest male participant level of any study was 50.5% (*n* = 2).

### Knowledge and Willingness

Few studies (*n* = 6) exhibit some understanding of the term sustainable diets. However, the understanding of the term sustainable diets seems to be improving as the years progress, with two of the six studies reporting consumer knowledge of sustainable diets published in 2022 and none published before 2017. Conversely, a large number of studies (*n* = 14) show that there appears to be a general lack of knowledge concerning what is meant by this term, with participants admitting to needing more information on this concept. Furthermore, several studies (*n* = 11) reported that some participants thought they understood sustainable diets; however, many communicated misconceptions (*n* = 3). For example, Mann et al. [20] found that most people believe that sustainable eating would only be slightly beneficial to the environment as they underestimated the environmental impact of farming, processing, and packaging. Likewise, three studies found that participants held cynical attitudes toward sustainability. Regarding misconceptions, three studies discovered that participants erroneously overstated the environmental impact of food transportation compared to other factors pertinent to sustainable diets.

General unawareness of the (un)sustainability of particular foods was a frequent finding (*n* = 12). Eleven studies expressed an underestimation of the environmental impact of meat, with four overestimating the environmental impact of plant protein sources. One study, in particular, found that less than a fifth of participants agreed that reducing meat, dairy, and egg consumption would mitigate climate change [21]. Or another where participants “mistakenly perceived the environmental impact of soy-based meat substitutes as similar to that of conventionally produced meat” [22] (p. 196). Moreover, two studies revealed how participants were sceptical of the importance of sustainable diets lessening environmental degradation, whilst one was dubious of the health impacts of new alternative protein sources [23]. With references to gender differences, five studies found that women had more knowledge surrounding sustainability, with one contrasting and reporting no gender differences regarding knowledge of sustainable diets [24].

In the studies that looked at the motives for choosing food, health was near or at the top of the list in all (*n* = 16). Other motives included taste (*n* = 6), price (*n* = 6), and convenience (*n* = 2). Interestingly, environmental reasons were the third most mentioned food choice motive in a study by Culliford and Bradbury [25]. However, this is outnumbered by the 16 studies that conveyed that consumers either do not consider environmental reasons when purchasing food or that it has little influence on their food choices, with the former motives, especially health, dominating reasons for food choice. Correspondingly, the barriers to sustainable eating mirror the already stated motives for food choice. The barrier mentioned most frequently was connected to either a lack of knowledge regarding sustainable diets (*n* = 10) or attachment to meat and its difficulty to reduce (*n* = 9). Another frequent barrier reported was that sustainable diets are perceived as expensive (*n* = 9). Other barriers related to a lack of skills concerning the preparation of sustainable meals (*n* = 5), limited availability of sustainable food (*n* = 4), high preparation/cooking time of sustainable protein sources, namely pulses (*n* = 4), and expectations that plant-based food is not tasty (*n* = 3).

Interestingly, the studies that used a Likert scale to assess consumer willingness to adopt sustainable diets (*n* = 3) all reported that consumers were more open than not to adopting a sustainable diet. Additional studies (*n* = 6) also indicated consumers’ positive motives towards adopting sustainable diets. For example, health reasons were the main motive for accepting the ‘Meatless Monday’ challenge in Ramsing et al. [26], but environmental reasons were the main motive for continuing the challenge. Nevertheless, of the studies exploring consumer willingness (*n* = 26), nine reported that consumers were willing to adopt a sustainable diet. However, these were outnumbered by studies reporting unwillingness (*n* = 17) to adopt sustainable diets. The main component of unwillingness was reluctance to reduce meat consumption (*n* = 9), followed by low intention to adopt a plant-based diet (*n* = 4) and low acceptance of alternative protein sources (*n* = 4). In addition, two studies reported a low willingness to adopt a sustainable diet in general. Moreover, whether knowledge of healthy, sustainable diets transferred to implementing a sustainable diet was contradictory among the studies. Five reported that knowledge regarding sustainable diets did not transfer to implementing a sustainable diet, whilst 11 found that there was a positive correlation between knowledge of sustainable diets and the implementation of a sustainable diet.

Regarding gender, females are more willing to either reduce meat consumption or adopt sustainable eating habits (*n* = 20), whilst one study [27] reported men to be more willing to adopt a sustainable diet and two stated how intentions to eat sustainably did not differ by gender. Additional study characteristics, namely study aim, data collection tool, sample size, and specific target group, can be seen below (Table 2).

## 4. Discussion

This scoping review aimed to complete a review of the studies conducted on consumers concerning their knowledge of a sustainable diet and their willingness to adopt a sustainable diet. From this, 45 articles were deemed eligible for this review. As illustrated in Figure 2, studies published in this area have steadily increased in the previous 10 years.

### 4.1. Trends in Research

Evidently, 60% of the selected studies were published between 2020 and 2022. The increase in publications central to this research area is likely due to the growing awareness of the impact of climate change. For instance, the popular EAT-Lancet Commission Report published in The Lancet in 2019 by Willett et al. [2] perhaps spurred a movement that galvanised the need for research in the area of consumer orientation pertaining to sustainable diets. In addition, the rampant growth of publications may be because sustainability is now an interdisciplinary approach that almost all sectors have adopted. Thus, the food industry is following suit and seeking the need for research in sustainable diets as vital to uphold sustainability. Similarly, research in sustainable diets can also bring the food system closer to achieving the Sustainable Development Goals (SDGs) by 2030. This initiative emphasises responsible consumer consumption and food security, encouraging academics to research how the food system can align with the SDGs [65]. Nonetheless, the spread of this research worldwide varies.

### 4.2. Geographical Focus

Noticeably, European countries appear to be the frontrunners in gathering data pertaining to consumer knowledge and willingness concerning the adoption of a sustainable diet, with 28 of the 39 studies that focused on a single country coming from European nations. This could be because of the environmental regulations imposed upon European Union (EU) countries. An example of this is the EU Green Deal, which aims to make the EU the first region in the world with a climate-neutral status by 2050 [66]. Another factor is the EU’s adoption of a Circular Economy Action Plan, which promotes sustainable production intending to reduce a product’s environmental impact throughout its lifecycle [67]. Interestingly, during the literature search, no studies, to the knowledge of the author, have been published in Ireland. However, since the literature search, Safefood [68] has published a series of studies pertinent to this research area. Additionally, the Climate and Health Alliance [69] has also published a position paper regarding transitioning Ireland to a healthy and sustainable food system. From this, the aspiration is that population knowledge concerning the need to shift to a more sustainable diet increases.

### 4.3. Knowledge

As reported in the results, knowledge pertaining to a sustainable diet was low (*n* = 6). However, as seen in Figure 2, the growing body of research on this subject may suggest that this topic is relatively new. Therefore, time may be needed for consumers to develop an understanding of what constitutes a sustainable diet. Interestingly, in the subject of Home Economics in Irish post-primary schools, the topic of sustainability is a major focus in the new National Council for Curriculum and Assessment (NCCA) Junior Cycle Home Economics course [70]. This focus can be seen with the element of “sustainable and responsible living” being integrated throughout the three strands of this subject’s curriculum [70] (p. 12). This is of particular importance as one of these strands is “Food health and culinary skills” [70] (p. 10). Thus, enabling Junior Cycle Home Economics students to learn food health and culinary skills from a sustainability perspective. This is something that is not explicitly addressed in the current NCCA Leaving Certificate Home Economics course [71]. Therefore, a reform of this course could implement elements of sustainability to bridge the gap between the Junior Cycle and Leaving Certificate to continue and progress the learning of practical food skills using a sustainable approach. This lack of knowledge also appears to have generated misconceptions surrounding sustainable diets.

### 4.4. Misconceptions

Three studies reported that participants overstated the effects of transport, particularly air-freighted food, on the environment [20,22,25]. Conversely, figures depict that only 0.16% of food is transported by air, whilst 58.97% is carried by water [6]. In addition, the level of scepticism reported in three studies exposed the reality that many consumers do not have sufficient knowledge about the necessity of transitioning to a sustainable diet [20,23,31]. Ergo, this lack of knowledge surrounding sustainable diets may translate to a lesser chance of sustainability being a factor affecting food choices.

### 4.5. Barriers to Adoption

In terms of factors affecting food choice, health was deemed the most or one of the most important in all of the 16 studies that posed this question. Cost (*n* = 6), taste (*n* = 6), and convenience (*n* = 2) were other popular motives affecting food choice. However, choosing foods for environmental purposes was much less favoured, as 16 studies indicated that environmental reasons do not influence food choice to a great extent. Perhaps due to the lack of environmental knowledge among consumers, coupled with their general scepticism surrounding sustainable diets, as aforementioned.

Likewise, the prominent barriers inhibiting sustainable eating were lack of knowledge regarding sustainable diets (*n* = 10), attachment to meat (*n* = 9), perceived cost (*n* = 9), and inadequate culinary skills—particularly surrounding meat substitutes (*n* = 5). Attachment to meat is of particular significance as livestock-derived food is responsible for 72–78% of total agricultural emissions [72], and high consumption of red and processed meats has been linked to adverse effects on human health [73]. Moreover, red meat and processed meats have been classified as group 2A (probably carcinogenic to humans) and group 1 (carcinogenic to humans), respectively [74]. Thus, a reduction in red and processed meat is beneficial for human and planetary health [75]. Interestingly, females tend to be more open to adopting a sustainable diet than men.

### 4.6. Gender Differences

Although overall willingness to adopt a sustainable diet was relatively low, female participants (*n* = 20) seemed to be more inclined than their male counterparts (*n* = 1) to adopt a sustainable diet. One reason females may be more open to adopting a sustainable diet could come down to gender perceptions. A study by Rozin et al. [76] found that individuals who preferred a beef-based diet were perceived as less feminine and more masculine than those who preferred a vegetable-based diet. Ergo, this stigma may explain one reason why females are more willing to adopt a sustainable diet than men. Since there is such a difference regarding genders, perhaps a tailored education campaign may be advantageous to balance a willingness to adopt a sustainable diet amongst males and females.

### 4.7. Limitations

The first limitation of this scoping review is that many of the studies employed different research methods. Therefore, interpreting results is challenging as many of the qualitative studies provided in-depth analysis, whilst some of the quantitative studies gave more concise responses. Another limitation of this review could be the superficial analysis and interpretation of the results of each study, although such analysis would be beyond the remit of a scoping review and best suited to a systematic review [17]. However, a systematic review would not suit the aim of this paper, as the aim of this paper was to conduct a scoping review to map the literature pertaining to consumer knowledge concerning sustainable diets and their willingness to adopt a sustainable diet. Ergo, a scoping review was the most suitable review type to map the pertinent literature.

### 4.8. Further Research

Many of the results from the included studies are transferrable and can be employed from an Irish perspective. However, further studies are needed to look at Irish consumer perceptions of the environmental and health impact of foods. This would address both Irish consumers’ knowledge and perceptions of foods in terms of their effects on human and planet health. Furthermore, although Safefood [68] conducted a series of studies on the Irish population pertaining to sustainable diets, they did not carry out a study that focused on young Irish consumers. Thus, to the author’s knowledge, no studies on young consumers in Ireland exist regarding sustainable diets. Therefore, a future study could look at the knowledge and perceptions of young consumers to discover their knowledge about sustainable diets and their willingness to adopt a sustainable diet.

## 5. Conclusions

The current food system needs reform to mitigate the ongoing disastrous effects on both human and planetary health. One way to alleviate this concern is the shift towards a sustainable diet. Thus, this scoping review set out to map the current literature relating to consumer knowledge concerning sustainable diets and their willingness to adopt a sustainable diet. Additionally, the consensus from the included studies suggests that consumers generally lack knowledge about food sustainability and, in the main, are unwilling to adopt a sustainable diet at this current time. Misconceptions surrounding sustainable diets exist, and this is perhaps due to a lack of information in this area. Therefore, more research-based initiatives in this area may encourage the transition towards a diet that promotes human health whilst also preserving the planet. Consequently, more research in this research area is required, specifically research conducted on young consumers to further the body of knowledge in this area.

## Figures and Tables

**Figure 1 nutrients-16-04254-f001:**
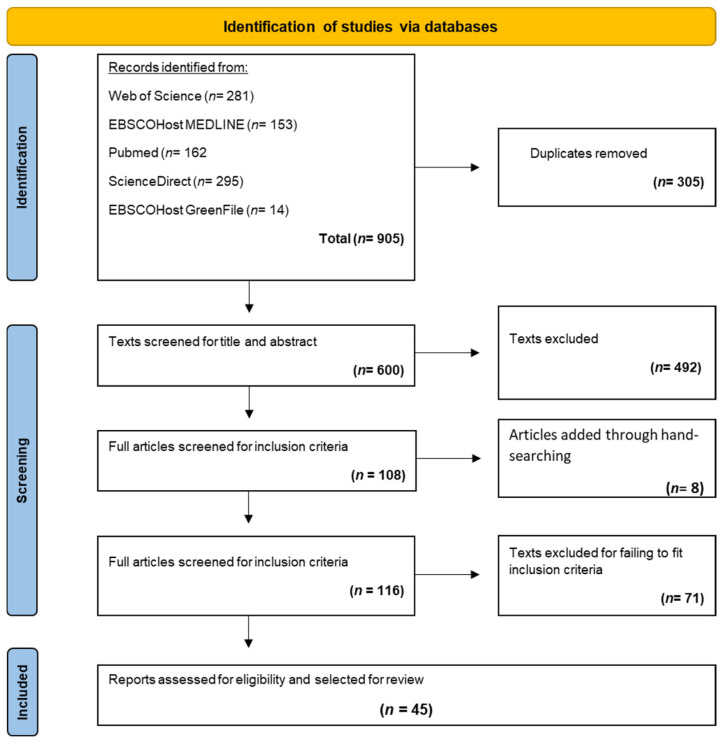
PRISMA flow chart of the article selection process.

**Figure 2 nutrients-16-04254-f002:**
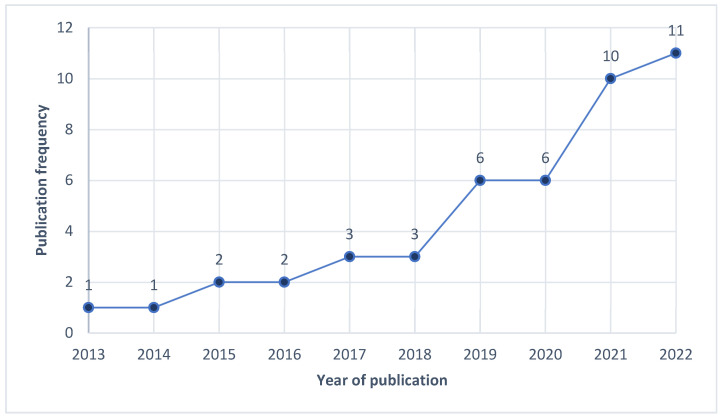
Publication year frequency (*n* = 45).

**Figure 3 nutrients-16-04254-f003:**
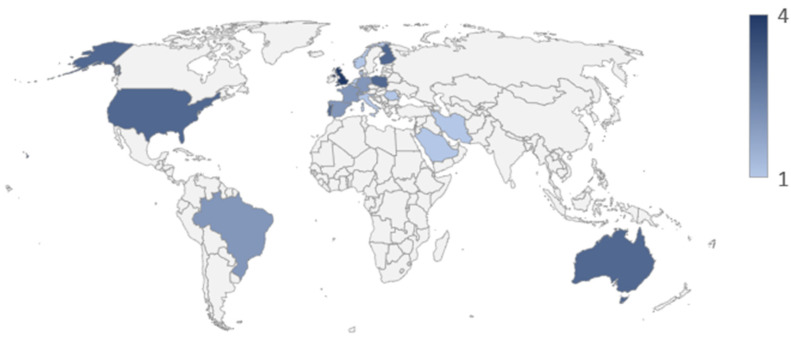
Demographic frequency (*n* = 39).

**Figure 4 nutrients-16-04254-f004:**
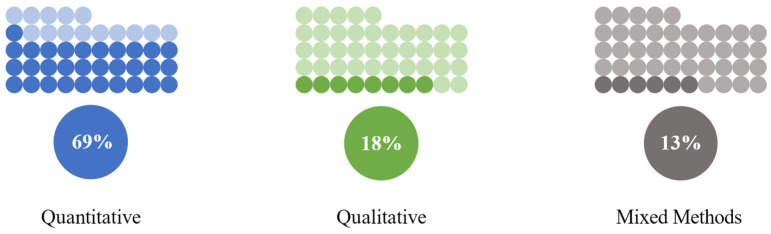
Methods frequency (*n* = 45).

**Table 1 nutrients-16-04254-t001:** Search terms derived from each concept.

Concept	Search Terms
Sustainable Diet	(“Sustainable diet *” OR “Ecological diet *” OR “Environmental diet *” OR “Planetary diet *”)
	AND
Knowledge/Willingness	(Willingness OR knowledge OR attitudes OR views OR opinions OR perspectives)

**Table 2 nutrients-16-04254-t002:** Characteristics of included articles (*n* = 45).

Reference	Author	Year	Country	Study Aim/Objective	Methods	Data Collection Tool	Sample Size	Target Group	Participant Gender (% Female)
[28]	Vanhonacker et al.	2013	Belgium (Flanders)	*“This study investigates opportunities and bottlenecks of some alternative and more sustainable food choices in terms of consumer evaluation.”* (p. 7)	Quantitative	Survey	221	Flanders resident 18 or over	64.3% female
[29]	de Boer et al.	2014	Netherlands	*“To provide insight into the underlying logic of the strategies and explore how they can work to change the amount and source of protein consumption, drawing on survey data from the Netherlands.”* (p. 120)	Quantitative	Survey	1083	Dutch residents 18 or over	50% female
[21]	Clonan et al.	2015	UK (England) (Nottinghamshire)	*“To investigate consumer’s self-reported red and processed meat consumption (from intake and purchasing data) against/towards animal welfare, human health and environmental sustainability.”* (p. 2448)	Quantitative	Survey	842	Nottinghamshire residents 18 or over	59% female
[30]	Graca et al.	2015	Portugal	*“This study explored how representations of meat, perceived impacts of meat, and rationales for changing/not changing habits emerge associated with willingness to adopt a more plant-based diet.”* (p. 83)	Qualitative	Survey	410	Portuguese social media users 18 or over	66.9% female
[31]	Macdiarmid et al.	2016	UK (Scotland)	*“To explore in-depth public views and perceptions of the environmental impact of food, awareness of the link between climate change and meat, and to gauge the public’s opinions about their willingness to eat less meat as part of a more sustainable diet.”* (p. 488)	Qualitative	Focus groups and Interviews	87	Scottish residents in high and low socio-economic areas (rural and urban) 18 or over	54% female
[32]	Jallinoja et al.	2016	Finland	*“This article explores plant protein consumption frequencies, future intentions to increase bean consumption, and the associations of frequent bean eating with socioeconomic factors and bean-related meanings, material issues, and competence.”* (p. 4)	Quantitative	Survey	1048	Finnish residents 15 or over	58% female
[33]	Alles et al.	2017	France	*“To investigate the relationships between food choice motives including sustainability during purchasing, and dietary patterns in a large sample of French adults from the NutriNet-Santé Study.”* (p. 2)	Quantitative	Survey and 3 24 h food records	31,842	French adults living in the French metropolitan area. 18 or over	79% female
[34]	Péneau et al.	2017	France	*“Aimed at investigating the existence of dilemmas between health and environmental motives when purchasing meat, fish and dairy products, at determining the sociodemographic profiles of individuals reporting dilemmas, and finally at comparing dietary quality of these individuals with those reporting no dilemma.”* (p. 2)	Quantitative	Survey and 3 24 h food records	22,935	French residents 18 or over	75.2% female
[35]	Van Loo et al.	2017	UK, Germany, Belgium, and Netherlands	*“To provide insight into consumers’ motivation to eat healthily and sustainably, as measured by involvement.”* (p. 48)	Quantitative	Survey	2783	A resident of one of the studied countries 18 or over	50.1% female
[36]	Asvatourian et al.	2018	UK (Scotland) (Southwest)	*“To identify dietary patterns and their associated GHG emissions, then to explore their relationship, as domain-specific behavioural patterns, with measures of environmental attitudes and behaviours.”* (p. 217)	Quantitative	2 surveys	422	Southwest Scottish residents 18 or over	55% female
[37]	Harray et al.	2018	Australia (Western Australia)	*“To determine the level of community concern about impacts on future food supplies and the perception of the importance placed on government regulation over the supply of environmentally friendly food and identify dietary and other factors associated with these beliefs.”* (p. 226)	Quantitative	Secondary data analysis ○ Survey	2832	Residents in Western Australia 18 or over	49.8% female
[20]	Mann et al.	2018	Australia (Victoria)	*“To gain an in-depth understanding of participants’ attitudes, knowledge, perceived effectiveness (a person’s belief that his/her behaviour can contribute to environmental preservation) and behaviours relating to a sustainable eating pattern.”* (p. 2714)	Qualitative	Interview	24	Victorian residents who are involved in their household’s food preparation. 18 or over	54% female
[38]	Lehikoinen and Salonen	2019	Finland	*“Focus on sustainable diets from citizens’ perspectives and study the potential transition towards more sustainable diets.”* (p. 4)	Mixed Methods	Survey	2052	Finnish citizens 18 or over	51.1% female
[22]	Siegrist and Hartmann	2019	Switzerland (German/French speaking region)	*“To examine how consumers assess the environmental impact of various foods and how these perceptions influence two strategies that consumers may choose to reduce the environmental impact of their food consumption, namely, consumption of meat substitutes and organic meat.”* (p. 196)	Quantitative	Secondary data analysis ○ Survey	5586	Residents from the German- and French-speaking parts of Switzerland. 18 or over	52% female
[39]	Graca et al.	2019	Portugal	*“To generate inputs for developing, marketing, and promoting plant-based meals and food products, using a consumption-focused approach.”* (p. 20)	Quantitative	Interview	1600	Portuguese residents 18 or over	52.6% female
[40]	Rejman et al.	2019	Poland (Mazovia)	*“To obtain better insight into the conceptualization of sustainable consumption by Polish consumers with special focus on food choice determinants relevant to the concept of sustainability.”* (p. 1332)	Quantitative	Survey	600	A resident of a city in Mazovia 18 or over	61.1% female
[41]	Barone et al.	2019	Brazil (Sao Paulo)	*“To investigate the relationship between sustainability and food, and other possible associations with the socio-demographic characteristics and consumer segmentation, as well as to identify the characteristics of sustainable and unsustainable foods and the sustainable diet concept from a consumer perspective.”* (p. 206)	Qualitative	Survey	150	Sao Paulo residents 18 or over	62% female
[42]	Larson et al.	2019	USA	*“To describe continuity over time in reports of valuing sustainable diet practices and investigate relationships between values, household meal behaviours, and dietary intake.”* (p. 2598)	Quantitative	Survey	1620	Young adults who participated in Project EAT in 2003–2004 and 2015–2016.	58% female
[25]	Culliford and Bardbury	2020	UK	*“To evaluate the perceived environmental benefit of sustainable dietary recommendations, readiness to adopt these behaviours, and differences in perceived importance and reported behaviours between demographic groups in a UK sample.”* (p. 4)	Quantitative	Survey	442	UK residents 18 or over	66.1% female
[43]	García-González et al.	2020	Spain	*“To evaluate the knowledge on food sustainability and environmental impact concepts, as well as related attitudes and behavior in a representative sample of Spanish adult population.”* (p. 2)	Quantitative	Survey	2052	Spanish resident 18 or over	57% female
[44]	Duarte et al.	2020	Portugal	*“To characterize current pulse consumption in a sample of adult Portuguese population and to describe the potential drivers and barriers to the inclusion of this source of protein in the Portuguese diet.”* (p. 2)	Mixed Methods	Survey and Interview	1174	Portuguese residents 18 or over	71.4% female
[45]	Voinea et al.	2020	Romania (Bucharest)	*“To provide solutions for reshaping the food pattern by incorporating the principles of a sustainable diet.”* (p. 6)	Qualitative	Interview	21	Romanian traditional food consumers 18 or over	76% female
[46]	Szczebylo et al.	2020	Poland	*“To investigate the attitudes towards pulses among Polish consumers and to understand the perceived opportunities and barriers to increasing consumption of these foods.”* (p. 2)	Quantitative	Survey	1027	Polish urban employees aged 25–40	52% female
[47]	Tepper et al.	2020	Israel	*“We explore Israeli young adults’ perceptions of food-related sustainability issues, which can be integrated into future policies.”* (p. 2)	Quantitative	Survey	348	Israeli adults aged 20–45	51.4% female
[26]	Ramsing et al.	2021	USA (New York)	*“To track quantitative changes in participants attitudes and beliefs around meat reduction and gauge frequency at which households participating in the Campaign consumed meat.”* (p. 379)	Mixed methods	Intervention and Surveys	468	Households in Bedford, New York 18 or over	85% female
[48]	Fink et al.	2021	Germany	*“To explore the external factors availability, education, advertising and price that can cause the emergence of an intention-behavior gap while people are trying to nourish themselves sustainably.”* (p. 15)	Mixed methods	Think aloud study, questionnaire, and follow-up interview	20	German resident 18 or over	65% female
[49]	Gonera et al.	2021	Norway	*“The present study investigated the ‘adoption of a plant-based diet and the reduction of meat consumption’ as an innovation.”* (p. 2)	Mixed methods	Focus groups, observations, interviews, and Survey	1785	Norwegian resident 18 or over	49.5% female
[50]	Haghighian Roudsari et al.	2021	Iran (Tehran)	*“To explore the components of a sustainable diet among the factors that affect people’s food choices.”* (p. 1)	Qualitative	Interviews	33	Tehran resident 30–64	66.67% female
[51]	Hielkema and Lund	2021	Denmark	*“This study examined public willingness to reduce meat consumption in Denmark, and the drivers and barriers involved.”* (p. 1)	Quantitative	Survey	1005	Danish residents 18 or older	59.7% female
[52]	Kirbis et al.	2021	Slovenia	*“To examine the prevalence of sustainable dietary eices and attitudes among the Slovenian public and to investigate the role of education in fostering sustainable dietary patterns.”* (p. 1)	Quantitative	Secondary data analysis ○ Interview	1079	Slovenian residents 18 or over	51.2% female
[53]	Smiglak-Krajewska and Wojciechowska-Solis	2021	Poland	*“To identify the motives and barriers which, according to a consumer, influence the level of consumption of pulses.”* (p. 1)	Quantitative	Survey	1067	Polish residents 18 or over	52.3% female
[24]	Polleau and Biermann	2021	Germany	*“This study examines consumers’ beliefs on sustainable diets and how these beliefs are related to scientific evidence on sustainable diets.”* (p. 2)	Quantitative	Survey	420	German speakers 18 or over	Not specified
[54]	Niva and Vainio	2021	Finland	*“To identify different groups of consumers based on self-reported changes in the consumption of beef and plant and insect-based protein products.”* (p. 2)	Quantitative	Survey	1000	Finnish residents living in mainland Finland. 18 or over	49.8% female
[23]	Fox et al.	2021	USA (Northern California, Southeastern Nebraska)	*“To explore the mental models involved in food decisions and identify challenges and opportunities involved in shifting towards healthy and sustainable diets.”* (p. 2)	Mixed methods	Interviews and pile-sorting survey	27	Residents of Northern California or Southeastern Nebraska 18 or over	63% female
[27]	Alnasser and Musallat	2022	Saudi Arabia	*“To assess the extent to which Saudi society is aware of food sustainability, the potential effects of their purchasing behaviours based on the goals of Vision 2030, and their willingness to change their dietary and purchasing behaviour.”* (p. 2)	Quantitative	Survey	398	Saudi Arabia resident 18 or over	62.8% female
[55]	Pucci et al.	2022	Italy, Germany, Poland, USA, Brazil, Japan, Korea, China	*“To explore which factors influence a person’s attitude towards adopting a sustainable diet.”* (p. 291)	Quantitative	Survey	5501	A resident of one of the studied countries 18 or over	57.5% female
[56]	Simeone and Scarpato	2022	Italy (Rome)	*“To analyse consumers’ perceptions of food sustainability and attitudes towards food consumption and the effect on the environment and their willingness to include insects as a sustainable solution to reduce meat consumption in their diet.”* (p. 3)	Quantitative	Survey	117	Individuals responsible for food purchases 18 or over	56.41% female
[57]	Lourenco et al.	2022	Brazil (Sao Paulo)	*“To investigate the effect of individuals’ predisposition to adopt sustainable diets and to reduce or exclude meat intake, along with the consequences of the perception of barriers in moderating this relationship.”* (p. 3)	Quantitative	Survey	497	Sao Paulo residents 18 or over	56% female
[58]	Gaspar et al.	2022	Spain (University of Barcelona)	*“To analyze the level of knowledge and perceptions of food sustainability in a university community from Spain.”* (p. 2)	Quantitative	Survey	1220	Individuals working or studying at the University of Barcelona. 18 or over	68.3% female
[59]	Varela et al.	2022	Norway, France	*“To better understand the perception of Norwegian and French consumers towards increased utilization of high protein plant-based foods, their underlying attitudes, the barriers, and opportunities and concrete needs with regards to products, in order to increase the likelihood of a shift in diet towards a more plant-based diet.”* (p. 2)	Qualitative	Creative focus groups	30	Norwegian and French consumers 18 or over	50% female
[60]	Baur et al.	2022	Switzerland	*“To explore the drivers and barriers toward healthy and environmentally sustainable eating.”* (p. 14)	Quantitative	Survey	620	Swiss residents 18 or over	52% female
[61]	Migliavada et al.	2022	Italy, Turkey	*“To explore the links between environmental sustainability knowledge, psychological traits, sociocultural differences, and eating behaviors.”* (p. 1)	Quantitative	Survey	1888	Italian or Turkish national 18 to 75 years old	65.3% female
[62]	de Boer and Aiking	2022	EU Countries	*“To provide a systematic, multivariate analysis of the position of meat reduction in consumers’ beliefs about what ‘eating a healthy and sustainable diet’ involves.”* (p. 3)	Quantitative	Secondary data analysis ○ Interview	27,237	A resident of one of the studied countries	Weighted for gender
[63]	Perez-Cueto et al.	2022	Austria, Denmark, France, Germany, Italy, Holland, Poland, Romania, Spain, and UK	*“To evaluate differences on consumers’ perceived barriers towards consuming plant-based foods by dietary lifestyles in 10 EU countries, and hence, further inform strategies for a sustained dietary shift.”* (p. 2)	Quantitative	Survey	7590	A resident of one of the studied countries 18 or over	49.5% female
[64]	Ronto et al.	2022	Australia	*“To explore young Australians’ perspectives, motivators, and current practices in achieving a sustainable and healthy diet.”* (p. 2958).	Qualitative	Interview	22	Australian adults aged 18–25	77% female

## Data Availability

The original contributions presented in this study are included in the article. Further inquiries can be directed to the corresponding author.

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
