# Peer review of "Consumer Knowledge and Willingness Pertaining to the Adoption of a Sustainable Diet: A Scoping Review"

_nutrients, 2024, doi:10.3390/nu16244254_

Round 1

Reviewer 1 Report

Comments and Suggestions for Authors

Dear Authors,

I assess the reviewed manuscript as correct, properly structured, and well-written, showing all the necessary methodological information. Above all, it is important in terms of content. This is because it turned out that only 45 papers met the search criteria, given the publicity that sustainability issues have received in the analyzed time range. Certainly, the collection of these publications in Table 2 will be useful for other researchers dealing with the transformation of diets to sustainability. In this context, the indicated directions for further research are appropriate.

It has already been shown many times that populations can change their eating habits. Therefore, studying the behavior, attitudes, and nutritional knowledge of consumers regarding the adoption of a planetary diet is crucial for the transformation of food systems. According to WWF (2020), dietary shifts are potentially the quickest action to achieve nature restoration. They can help facilitate the other two actions, namely reduction in food loss and waste and the adoption of nature-positive agricultural and food processing practices.

Nevertheless, I have a few minor suggestions:

On page 6, in the paragraph on consumer willingness to adopt a sustainable diet, there is no information on how many publications have studied this issue (I counted 26). The sentence in L. 183-184 is superfluous since the same thing is written in the preceding sentence.

The title of Fig. 3 should state n = 39 since multi-country studies were not included.

Table 2 - three suggestions:

1- it is redundant to use the top reference (asterisk) at publication #54 and then explain it below the table. This information adds nothing to the substance of the issue,  

2- it is worth using the chronological order of publications in the table, since earlier (Fig. 2) and later (L. 5 Disc.) it was pointed out that the number of publications increases over time,

3- in addition, it is worth including in one of the columns a symbolic designation of publications on knowledge (K) and willingness (W) to adopt a sustainable diet. 

L. 89-90 (Concl.) - I don't see the need to cite 2 references (only two) in this sentence since the transformation of food systems to sustainable ones is an understandable challenge for both academia and business.

L. 100 (Concl.) - why only among young consumers in Ireland? FAO's new initiative Food Systems Youth Leadership Programme aims to empower young leaders to transform global food systems for a resilient, sustainable, and equitable future”.

Kind regards

KR

Author Response

Dear Reviewer, 1,

We would like to sincerely thank you for taking the time to review the article, “Consumer Knowledge and Willingness Pertaining to the Adoption of a Sustainable Diet: A Scoping Review.” We deeply appreciate your thoughtful feedback, insightful suggestions, and the valuable time you invested in reviewing this article.

We have attached the response to each comment, alongside the amendment of each suggested change. We have also uploaded the revised article, with your suggestions in place, to the online portal.

We are grateful for your support in improving our work. We look forward to hearing back from you and receiving feedback from you again.

Kind Regards,
Connor Dupuits, Elaine Mooney, Amanda McCloat

Comment 1: On page 6, in the paragraph on consumer willingness to adopt a sustainable diet, there is no information on how many publications have studied this issue (I counted 26).

Response 1: Thank you for spotting this. You are right! There are 26 articles which alluded to this issue.

Amendment: “Nevertheless, of the studies exploring consumer willingness (n = 26), nine reported that consumers were willing to adopt a sustainable diet. However, these were outnumbered by studies reporting unwillingness (n = 17) to adopt sustainable diets.”

(P. 6, L. 189-192).

Comment 2: The sentence in L. 183-184 is superfluous since the same thing is written in the preceding sentence.

Response 2: Thank you for this. Correct, no need for this sentence. This has been removed from the article.

Amendment 2: This sentence has been removed.

Comment 3: The title of Fig. 3 should state n = 39 since multi-country studies were not included.

Response 3: Thanks for pointing this out, it makes sense.

Amendment 3: We have updated the manuscript to n=39 countries.

(p. 4, L. 127).

Comment 4: 1- it is redundant to use the top reference (asterisk) at publication #54 and then explain it below the table. This information adds nothing to the substance of the issue.

Response 4: We appreciate you bringing this to our attention. This has been updated.

Amendment 4: We have removed the asterisk and deleted the and the information below the table.

(p. 19-20).

Comment 5: It is worth using the chronological order of publications in the table, since earlier (Fig. 2) and later (L. 5 Disc.) it was pointed out that the number of publications increases over time.

Response 5: This makes a lot of sense; We are grateful you pointed this out!

Amendment 5: We have put all articles in this table in chronological order.

(p. 7-20.)

Comment 6: In addition, it is worth including in one of the columns a symbolic designation of publications on knowledge (K) and willingness (W) to adopt a sustainable diet. 

Response: I am unsure of what you mean here. Do I add a new column to Table 2 and call it (“Knowledge (K)” and “Willingness (W)”) and put either a “K” or “W” or “K & W” that is relevant to each corresponding article?

Comment 7: I don’t see the need to cite 2 references (only two) in this sentence since the transformation of food systems to sustainable ones is an understandable challenge for both academia and business.

Response 7: We appreciate you exhibiting this to us. We have removed both references from the conclusion.

Amendment 7: References removed from conclusion.

(P. 3 (Conclusion), L. 115).

Comment 8: Why only among young consumers in Ireland? FAO’s new initiative Food Systems Youth Leadership Programme aims to empower young leaders to transform global food systems for a resilient, sustainable, and equitable future”.

Response 8: Thank you for the comment. It is not specific to Ireland as many countries do not have research pertaining to the adoption of a sustainable diet.

Amendment 8: The word “Ireland” has been removed and we have kept the word “young.”

(p. 3 (Conclusion), L. 123).

Reviewer 2 Report

Comments and Suggestions for Authors

Dear authors,

The abstract is well-written, but I recommend that the authors pay more attention to their wording. For example, the word "Ireland" is repeated twice in close proximity within the text. This could be replaced, for instance, with "in this country": During the time of the literature search, no studies were conducted in Ireland. Research needs to be conducted in this country, specifically on young consumers, to explore their knowledge and willingness to adopt a sustainable diet.

The introduction is good, but please consider the following suggestions for imptoving it: 

1. Consider making the initial statement more engaging by connecting the population growth directly to the need for sustainable food systems. For example, “With the global population projected to reach nearly 10 billion by 2050, addressing the sustainability of our food systems is more critical than ever.”

2. Ensure that citation styles are consistent (e.g., [1] vs. [2] (p. 461)). 

3. Add brief transitions between paragraphs to guide the reader and strengthen the connection between the issues presented. For example, after mentioning environmental impacts, a sentence like, “The strain on natural resources is matched by growing concerns over public health outcomes.”

4. Clarify the aim of the scoping review in the last paragraph to make it more specific.

The methodology is well written, and the use of an established method from the literature to conduct a scoping review is commendable.

In my opinion, this section should also belong to methodology part: "Database searches yielded 905 articles which were uploaded to the referencing man-103 agement software Zotero. These were screened for title and abstract per the eligibility cri-104 teria. Hence, 116 articles were uploaded to Rayyan, a research collaboration software [19]. 105 After examining the articles uploaded to Rayyan, two independent researchers were in-106 vited to review the articles and 45 articles were deemed acceptable to be included in the 107 review (Figure 1). "

In addition, please specify: how the title and abstract screening was conducted?

Please, mention what specific criteria were used to include or exclude studies during the screening in Rayyan. Briefly explain how Rayyan facilitates the screening process for those unfamiliar with it. Mention how disagreements were handled between the two researchers, if applicable.

The results are nice presented. In the discussion section, I suggest to divide the discussion into clear subsections (e.g., "Trends in Research," "Geographical Focus," "Barriers to Adoption," "Gender Differences") and to use linking sentences.

Explain the significance of statistical findings (e.g., the rise in publications) and provide context for regional trends. Highlight why European countries dominate this research and identify contributing factors like policy or cultural attitudes. Provide more insight into why females may be more inclined to adopt sustainable diets, exploring social or psychological factors. Propose  educational campaigns to encourage greater male participation, suggesting practical implementations.

Author Response

Dear Reviewer, 2,

We would like to sincerely thank you for taking the time to review our article, “Consumer Knowledge and Willingness Pertaining to the Adoption of a Sustainable Diet: A Scoping Review.” We deeply appreciate your thoughtful feedback, insightful suggestions, and the valuable time you invested in reviewing this article.

We have attached the response to each comment, alongside the amendment of each suggested change. We have also uploaded the revised article, with your suggestions in place, to the online portal.

We truly appreciate your support in enhancing our work. We look forward to your response and to receiving your feedback once again.

Kind Regards,
Connor Dupuits, Elaine Mooney, Amanda McCloat

Comment 1: I recommend that the authors pay more attention to their wording. For example, the word "Ireland" is repeated twice in close proximity within the text. This could be replaced, for instance, with "in this country": During the time of the literature search, no studies were conducted in Ireland. Research needs to be conducted in this country, specifically on young consumers, to explore their knowledge and willingness to adopt a sustainable diet.

Response 1: Thank you very much for pointing this out to me. This makes complete sense, and We have updated the abstract accordingly.

Amendment 1: “During the time of the literature search, no studies were conducted in Ireland. Research needs to be conducted in this country, specifically on young consumers, to explore their knowledge and willingness to adopt a sustainable diet.”

(p. 1, L. 18-20).

Comment 2: Consider making the initial statement more engaging by connecting the population growth directly to the need for sustainable food systems. For example, “With the global population projected to reach nearly 10 billion by 2050, addressing the sustainability of our food systems is more critical than ever.”

Response 2: We appreciate you stating this, along with your suggestion to include it in the article. We have amended the article as per your suggestion.

Amendment 2: “As the global population is projected to approach 10 billion by 2050, ensuring the sustainability of our food systems is more essential than ever.”

(p. 1, L. 25-26.)

Comment 3: Ensure that citation styles are consistent (e.g., [1] vs. [2] (p. 461)).

Response 3: Thank you for this comment. However, I am unsure what you mean, I used the prescribed MS Word template while completing this article and this is how I interpreted how to reference a quote. Can you provide me with more information so I can apply this suggestion? Thanks.

Comment 4: Add brief transitions between paragraphs to guide the reader and strengthen the connection between the issues presented. For example, after mentioning environmental impacts, a sentence like, “The strain on natural resources is matched by growing concerns over public health outcomes.”

Response 4: We are grateful that you pointed this out. We have updated the paragraphs to make them more coherent.

Amendment 4: “In addition, this strain on the environment is mirrored by increasing public health concerns.”

(p. 1, L. 33-34.)

Comment 5: Clarify the aim of the scoping review in the last paragraph to make it more specific.

Response 5: Thank you for suggesting this. We have made the aim more concise to provide more clarity.

Amendment 5: “This paper aims to conduct a scoping review of the literature pertaining to consumer knowledge concerning a sustainable diet and their willingness to adopt a sustainable diet.”

(p. 2, L. 58-59).

Comment 6: In my opinion, this section should also belong to the methodology part: "Database searches yielded 905 articles which were uploaded to the referencing man-103 management software Zotero. These were screened for title and abstract per the eligibility criteria. Hence, 116 articles were uploaded to Rayyan, a research collaboration software [19]. 105 After examining the articles uploaded to Rayyan, two independent researchers were invited to review the articles and 45 articles were deemed acceptable to be included in the review (Figure 1). "

Response 6: You are right! This is best suited to the methodology. Thus, we have moved this from the results section to the methodology.

Amendment 6: Moved to the methodology section.

(p. 4, L. 101-115.)

Comment 7: In addition, please specify: how the title and abstract screening was conducted?

Response 7: Thank you for the comment. We agree that more detail is needed regarding the information on screening.

Amendment 7: We have provided information on how title and abstract screening was conducted.

“These were initially screened for title and abstract per the eligibility criteria. Articles passed initial screening if their title contained words or sentences included in the search terms derived from each concept (“sustainable diet” and “knowledge/willingness”) (Table. 1). The abstracts of articles that passed screening by title were then screened in more detail. Articles progressed to the next stage of selection if they met the eligibility criteria aforementioned”

(p. 4, L.102-107).

Comment 8: Please, mention what specific criteria were used to include or exclude studies during the screening in Rayyan.

Response 8: We appreciate this suggestion, and we have provided details regarding the same.

Amendment 8: “Articles progressed to the next stage of selection if they met the eligibility criteria aforementioned. Hence, 116 articles were uploaded to Rayyan, a research collaboration software.”

(p. 4, L. 106-108).

“The methods and results of these articles were reviewed for eligibility as per inclusion criteria.”

(p. 4, L. 111-112)

Comment 9: Briefly explain how Rayyan facilitates the screening process for those unfamiliar with it. Mention how disagreements were handled between the two researchers, if applicable.

Response 9: This is something I should have included so thank you for highlighting this to me.

Amendment 9: “Rayaan enables researchers to screen articles based on their eligibility criteria. The invited researchers can select articles they wish to include, exclude, or are undecided about. Any disagreements are highlighted, and the researchers can discuss these to decide whether to include or exclude these articles.”

(p. 4, L.108-111).

“Any disagreements among articles were resolved by the second independent reviewer.”

(P. 4, L.115-116).

Comment 10: In the discussion section, I suggest to divide the discussion into clear subsections (e.g., "Trends in Research," "Geographical Focus," "Barriers to Adoption," "Gender Differences") and to use linking sentences.

Response 10: Thank you for this suggestion! This certainly makes the article more coherent and allows the paragraphs to flow.

Amendment 10: We have added subheadings to each point discussed in the discussion. We have also added linking sentences at the end of each paragraph.

(P. 1-2 (Discussion), L. 1-112).

Comment 11: Explain the significance of statistical findings (e.g., the rise in publications) and provide context for regional trends. Highlight why European countries dominate this research and identify contributing factors like policy or cultural attitudes. Provide more insight into why females may be more inclined to adopt sustainable diets, exploring social or psychological factors. Propose educational campaigns to encourage greater male participation, suggesting practical implementations.

Response 11: We appreciate the comments and suggestions for developing the discussion. We have updated the discussion as per your suggestions. I

Amendment 11: We have updated the discussion as per your suggestions.

“In addition, the rampant growth of publications may be because sustainably is now an interdisciplinary approach that almost all sectors have adopted. Thus, the food industry is following suit and seeking the need for research in sustainable diets as vital to uphold sustainability. Similarly, research in sustainable diets can also bring the food system closer to achieving the Sustainable Development Goals (SDGs) by 2030. This initiative emphasizes responsible consumer consumption and food security, encouraging academics to research how the food system can align with the SDGs [73]. Nonetheless, the spread of this research worldwide varies.”

(p. 1 (Discussion), L. 12-19)

“A reason for this could be because of the environmental regulations imposed upon European Union (EU) countries. An example of this is the EU Green Deal which aims to make the EU the first region in the world with a climate-neutral status by 2050 [74]. Another factor is the EU adoption of a Circular Economy Action Plan which promotes sustainable production intending to reduce a product's environmental impact throughout its lifecycle [75].”

(p. 1 (Discussion), L. 24-29)

“One reason females may be more open to adopting a sustainable diet could come down to gender perceptions. A study by Rozin et al. [76] found that individuals who preferred a beef-based diet were perceived as less feminine and more masculine than those who preferred a vegetable-based diet. Ergo, this stigma may explain one reason why females are more willing to adopt a sustainable diet than men.”

(p. 2 (Discussion), L. 82-86).

Round 2

Reviewer 2 Report

Comments and Suggestions for Authors

Dear authors,

The paper was significantly improved according to reviewers' suggestions.

Congratulations!